# A Synergistic Overview between Microfluidics and Numerical Research for Vascular Flow and Pathological Investigations

**DOI:** 10.3390/s24185872

**Published:** 2024-09-10

**Authors:** Ahmed Abrar Shayor, Md. Emamul Kabir, Md. Sartaj Ahamed Rifath, Adib Bin Rashid, Kwang W. Oh

**Affiliations:** 1Department of Mechanical Engineering, Khulna University of Engineering & Technology, Khulna 9203, Bangladesh; shayorme16kuet@gmail.com (A.A.S.); sartaj.rifath789@gmail.com (M.S.A.R.); 2Sensors and MicroActuators Learning Lab (SMALL), Department of Electrical Engineering, The State University of New York at Buffalo, Buffalo, NY 14260, USA; 3Department of Industrial and Production Engineering, Military Institute of Science and Technology, Dhaka 1216, Bangladesh; 4Department of Biomedical Engineering, The State University of New York at Buffalo, Buffalo, NY 14260, USA

**Keywords:** vascular flow, microfluidics, device fabrication, blood flow analysis, hematological disease, computational fluid dynamics (CFDs)

## Abstract

Vascular diseases are widespread, and sometimes such life-threatening medical disorders cause abnormal blood flow, blood particle damage, changes to flow dynamics, restricted blood flow, and other adverse effects. The study of vascular flow is crucial in clinical practice because it can shed light on the causes of stenosis, aneurysm, blood cancer, and many other such diseases, and guide the development of novel treatments and interventions. Microfluidics and computational fluid dynamics (CFDs) are two of the most promising new tools for investigating these phenomena. When compared to conventional experimental methods, microfluidics offers many benefits, including lower costs, smaller sample quantities, and increased control over fluid flow and parameters. In this paper, we address the strengths and weaknesses of computational and experimental approaches utilizing microfluidic devices to investigate the rheological properties of blood, the forces of action causing diseases related to cardiology, provide an overview of the models and methodologies of experiments, and the fabrication of devices utilized in these types of research, and portray the results achieved and their applications. We also discuss how these results can inform clinical practice and where future research should go. Overall, it provides insights into why a combination of both CFDs, and experimental methods can give even more detailed information on disease mechanisms recreated on a microfluidic platform, replicating the original biological system and aiding in developing the device or chip itself.

## 1. Introduction

The demand for more reliable and effective preclinical models has increased over time in an effort to minimize the use of animal models for drug efficacy and safety assessments [1]. While animal models share similarities with human cells and organs, they often fall short in accurately predicting human metabolism and behavior, in addition to being labor-intensive, ethically concerning, and costly. To address the limitations of in vivo testing, researchers have turned to advanced microfluidic devices that can integrate bio-models to mimic the physiology and function of cells and organs [2,3]. These microfluidic devices, which are the basic components of the microfluidic chip family, are characterized by microchannels with dimensions ranging from tens to hundreds of micrometers, enabling the processing of small volumes of fluids. Initially developed for analytical purposes, microfluidic devices have gained popularity due to their ability to operate with minimal reagents and samples, offering advantages such as rapid analysis, reduced fabrication and reagent expenses, miniaturization, selectivity, sensitivity, portability, and biocompatibility [4,5].

Miniaturized microscale devices are valuable tools for performing processes such as reactions, separations, or chemical identification [6]. Microfluidic devices may be referred to as microreactors, lab-on-a-chip, or organ-on-a-chip, depending on their specific application and functional characteristics, as mentioned in the literature [7,8,9,10,11,12]. Microfluidic chips may be made using several manufacturing processes and a wide variety of materials, depending on their intended purpose [10]. With the introduction of various production methods in the literature and their subsequent use in practice [6,13], the potential for advancements in the area of microfluidics is rapidly expanding, offering new opportunities for both the academic and industrial sectors [11]. In addition, this technology shows great potential for everyday use, as numerous commercially available devices are already being used for various purposes such as at-home pregnancy testing, rapid testing for viruses like HIV, COVID-19, Herpes Simplex, and Hepatitis A, B, and C, as well as blood glucose monitoring [11,12].

In addition to the advancements in the creation of novel microphysiological systems, computational tools have significantly contributed. The creation of these microfluidic devices is a complex and laborious procedure that often requires numerous costly and time-consuming design iterations [14]. To enhance the development of microfluidic devices, it is essential to use dependable and efficient simulation techniques to accurately forecast the functionality of a design. These tools facilitate the supplementation of experimental investigations, from an engineering perspective, enabling the examination of physical phenomena, evaluation of device feasibility, and enhancement of design efficiency at a reduced expense [15,16,17]. In addition, numerical approaches can identify crucial parameters, such as pressure, velocity, shear rate, and temperature, which are difficult to quantify using experimental techniques. To enhance their findings, scholars have supplemented their studies using numerical approaches. To model the flow of Newtonian fluid via a single microchannel, the Hagen–Poiseuille law [18] may be used. The flow profile throughout the channel is assumed to follow a parabolic shape [19]. Channel-based microfluidic systems are composed of several linked microchannels, necessitating the use of more comprehensive modeling techniques. In order to achieve this objective, the majority of designers use techniques from computational fluid dynamics (CFDs) [20], such as the finite volume method (FVM), finite element method (FEM) [21], or the Lattice Boltzmann Method (LBM) [22] to produce accurate findings. The latest surveys on microfluidics modeling have identified the finite volume method (FVM) [23], finite element method (FEM) [23], and Lattice Boltzmann Method (LBM) [24] as numerical techniques for solving the Navier–Stokes Equations (NSEs) in microfluidics. Therefore, these simulation approaches may be regarded as cutting-edge for simulating microfluidic devices.

In this review paper, we will analyze vascular flow using computational and experimental methods, with a particular emphasis on microfluidics as the method of investigation. We will begin by providing an overview of the various models and devices used for computational and experimental investigations. To this end, the many applications of using CFDs alone for researching diseases in cardiovascular systems and the experimental use of fabricated devices for the same purposes will be presented in this work. After that, we will explore the benefits and limitations of each method where possible. To conclude, we will discuss the relevance of these findings in clinical practice as well as the paths in which future research should. Table 1 provides some blood cell properties which have been used in much research mentioned here as standard and contrasted with to determine pathological conditions.

This review paper aims to contribute to a better understanding of the underlying mechanisms of diseases and blood flow in general and to inform the development of new treatments and interventions for these widely prevalent and serious medical conditions. This will be accomplished by providing a comprehensive overview of the current state of research on the application of microfluidics in a wide range of flow conditions and blood corpuscle movement analysis considering pathological conditions.

Being half of the total volume of blood, RBCs are the main factor influencing blood flow behavior in microcirculation. They are crucial in providing oxygen to all tissues, including those with the tiniest capillaries. RBCs are generated in the bone marrow, where they mature and lose their nuclei before leaving the marrow and entering the bloodstream. A healthy person’s hematocrit, or volume concentration of RBCs, is around 45%, and it has a significant impact on the behavior of blood rheology [28]. RBCs have a biconcave discoid shape while suspended in plasma, with a main diameter of roughly 8 µm and a thickness of 2 µm.

Nucleated WBCs are the body’s primary line of defense against infections, although they only make up around 1% of the entire volume of normal blood. They tend to be round, although their exterior is not always smooth. WBCs have a lower density than RBCs and a diameter ranging from around 6 to 20 m [29]. White blood cells (WBCs) play a crucial role in the immune system by fighting off harmful microorganisms such as viruses, bacteria, parasites, and fungi. Five different types of white blood cells (WBCs) exist: lymphocytes, monocytes, eosinophils, basophils, and neutrophils [30]. The presence or absence of WBCs in the blood can be used as a diagnostic tool for disorders such as leukemia, AIDS, autoimmune diseases, immunological deficiencies, and blood diseases [31].

## 2. Microfluidic Model of Blood Vessels

A major advancement in bioanalytical chemistry is the development of organ-on-a-chip technology, which models organs like the liver and lungs for bioassays and drug formulation. Blood vessels are also frequently targeted in these assays. The circulatory system, including blood vessels, delivers oxygen and nutrients to tissues and removes waste. The left ventricle pumps oxygenated blood through the aorta, arterioles, and capillaries. Arteries distribute blood, while capillaries facilitate material exchange. Deoxygenated blood returns to the heart’s right atrium via veins.

In 1840, while studying the flow of water via a glass tube, the law of Hagen–Poiseuille established the groundwork for fluid mechanics and hemorheology, which in turn gave rise to the microfluidic model [32]. A more accurate representation of the natural vessels can be achieved using better-designed surfaces, geometries, branching structures, and substrate materials. Using semicircular (rigid) glass plates with bifurcated channels, Cokelet [33] was one of the first to create a microvascular blood artery network using microsystems technology. Nearly ninety years ago, Murray [34] made the observation that, as Murray’s law goes, the sum of the cubes of the diameters of the daughter vessels is equal to the cube of the diameter of the parent vessels. A design of the chip that complies with Murray’s law gives uniform and steady flow rates in all channels, as Lim [35] showed. The impact of microchannel material on RBC deformability was studied by Tomaiuolo [36]. The local deformations in the channel walls are caused by cell–wall interactions in PDMS, which has a lower Young’s modulus than glass. In contrast, glass capillaries of the same diameter do not undergo any changes. Another model for the supramolecular layer (glycocalyx) that lines the inside of blood arteries can be constructed using polymer brushes attached to the walls of microcapillaries [37]. While lateral velocity was unaffected by brush thickness, RBC deformation did rise. Estrada [38] verified this by using a thin and flexible PDMS membrane to account for the blood vessel walls’ elasticity. Reconstructing the steps of vascular formation, such as anastomosis, sprouting angiogenesis, EC lining, and vasculogenesis, was made possible using a microfluidic network, according to Wang and colleagues [39]. Wang’s device design features side channels acting as arteries and veins, connecting to core tissue chambers. Nutrient availability can be easily adjusted. Optical photolithography creates microchannels with blood vessel dimensions, but replicating biomechanical characteristics and cellular interactions remains challenging. Research into both basic science and disease can make use of microfluidic platforms for red blood cell (RBC) studies based on deformability. Red blood cells (RBCs) deformed and passed through microchannels under controlled pressures, aiding in malaria detection and deformation evaluation. Techniques like magnetic sorting and incremental filtration helped analyze cell density and sort white blood cells (WBCs) based on their dynamics, mentioned in the Figure 1 [36,40,41,42,43,44,45].

## 3. Overview of CFDs for Vascular Flow

Computational fluid dynamics (CFDs) are widely used in studies for detecting biomarkers, diagnosing diseases, understanding drug efficacy, and analyzing the human body. They help design microfluidic devices by predicting outcomes and improving designs. This review focuses on the cardiovascular system, where blood flow is modeled using CFDs. Various “Lab-on-a-Chip” devices mimicking blood vessels have been designed with CFDs.

The review discusses methods for modeling cells and vehicles in flow, which are generally mesoscopic due to computational constraints. These methods are categorized into three types: (1) fluid flow modeling, (2) deformable cells modeling, and (3) coupled fluid flow and cell deformation modeling as show in Figure 2.

Most vascular CFD modeling studies follow a similar process to quantify hemodynamics from imaging data. This involves (1) identifying key vasculature from volumetric imaging, (2) creating a model from the identified vasculature, (3) generating a volume mesh of the model, and (4) running a CFD solver with suitable physiological boundary conditions based on observations and assumptions. Post-processing details are described in the previously reported works [46,47].

Depending on the study objectives, different software packages can be used to solve the governing differential equations of blood flow and vessel wall deformation. Commonly used packages include finite volume methods like Ansys Fluent, CFX, Siemens STAR CCM+, and OpenFoam, and finite element methods such as COMSOL, SimVascular, and Crimson. For structural wall stress analysis, FEA tools like Ansys Mechanical, Simulia Abacus, and SimVascular are utilized. The Arbitrary Lagrangian–Eulerian (ALE) formulation is implemented in Adina, SimVascular, and Simulia Abacus, while coupled CFD and FEA solvers for FSI are available in Ansys Workbench. When the focus is on hemodynamics rather than detailed structural stress and strain distributions, the coupled momentum method serves as an efficient alternative to the ALE formulation [47]. The accuracy of simulations depends on several factors, including patient-specific geometry, the chosen CFD methodology, blood properties, and flow rates at both the inlet and outlet [48]. These elements are crucial for ensuring that the simulations accurately reflect real-world conditions and provide reliable results.

Balancing computational effort and accuracy in blood flow simulations requires careful consideration of factors such as high-order numerical schemes, appropriate time-step sizes, and mesh sizes [49]. While laminar flow is typically assumed for physiological blood flow, turbulence can occur under various conditions, such as in the aorta with valve stenosis or stenotic arteries. Flow instability is revealed only under high spatial and time resolutions. For turbulent blood flow studies, different models are used: Reynolds-averaged Navier–Stokes (RANS) models [50], large eddy simulation (LES) [51], and direct numerical simulation (DNS) [52]. LES resolves large-scale velocity fluctuations and models laminar, transitional, and turbulent features, while DNS directly solves Navier–Stokes equations for all turbulent scales but requires significantly more computational resources.

However, machine learning (ML) algorithms trained on CFD and FEA data can significantly speed up these processes with less error and show excellent correlation with CFDs. For instance, deep neural networks (DNNs) can predict velocity and pressure fields in the thoracic aorta with minimal error in seconds [53]. Despite challenges in training data-driven models due to patient variability, advancements in ML-accelerated simulations hold promise for real-time clinical applications in biomechanics. These methods aid in imaging, image segmentation, estimating constitutive parameters, and predicting hemodynamics and stresses within vascular structures. Deep learning, particularly convolutional neural networks (CNNs) like the most popular U-Net [54], E-Net, and V-Net, has been effective in medical image analysis, enabling quick and unbiased vascular segmentation [55].

### 3.1. Blood as a Newtonian Fluid through Vessels

Blood is mainly a non-Newtonian fluid, but it is often simplified as Newtonian for easier analysis. Microfluidic models help study blood flow and develop diagnostic devices. However, at small scales, blood flow is not always continuous, and hematocrits must be measured for accuracy. Research shows that using a Newtonian model for microfluidic cases can lead to significant variations, indicating that it is an oversimplification. Additionally, considering only shear viscosity for blood analog solutions is insufficient [56].

In a study on metastasis, where tumor cells spread through blood vessels, blood was considered Newtonian to calculate wall shear stress (WSS). This involved injecting single circulating tumor cells (CTCs), observing CTC cluster disintegration due to shear stress, and reinjecting CTC clusters.

Thomas Glatzel et al. used CFDs for various microfluidic applications, including mixers, rotating platforms, and bubble dynamics studies, all with Newtonian fluids. The solvers worked well for convection–diffusion problems but were less reliable for free surface and surface tension issues, which are not relevant to blood flow studies [57].

CFDs and microfluidics are used to study blood as a Newtonian fluid in fields like hemolysis, where RBCs release hemoglobin, affecting clinical test accuracy. This deformation is due to small dimensions in microfluidics. In lab-on-a-chip systems for cell plasma segregation and identification, predicting hemolysis is crucial. The goal is to develop a low-cost numerical tool for biologically integrated microfluidic systems. RBCs are modeled as rigid spheres with viscoelastic properties, using Resolved CFD-DEM and an immersed boundary concept to simulate cell and fluid mechanics, aiding hemolysis prediction [58].

#### Blood Viscosity

Blood viscosity varies under different conditions, requiring careful analysis, especially for disease-related scenarios. CFD analysis is essential for device design. Nhut Tran-Minh and Frank Karlsen’s study examined blood flow in a passive planar micromixer using various viscosity models in COMSOL Multiphysics. This is crucial for drug delivery simulations, as blood viscosity changes with shear rate. A non-Newtonian model provides the most accurate representation. The study used a passive planar micromixer with elliptical micropillars to achieve efficient mixing in the laminar flow regime [59].

In a double Y-type microfluidic device, the flow and diffusion behavior of two viscous liquids were studied using CFDs. The relationship between viscosity ratio, solution concentration, and flow rate at the device’s exit was evaluated. When two viscous fluids were introduced at the same flow rate, the higher-viscosity fluid moved towards the lower-viscosity side. Increasing the flow rate of the low-viscosity fluid can mitigate the viscosity ratio’s impact on velocity and concentration distribution, as shown by simulations [60].

Dmitry A. Fedosov et al. used molecular dynamics and red blood cell models to study blood viscosity, predicting its impact on shear rate and hematocrit. They linked red blood cell behavior to non-Newtonian properties, suggesting future predictions of blood viscosity abnormalities in diseases like diabetes, AIDS, and malaria, depicted in Figure 3. Figure 3c illustrated the relative viscosity (the viscosity of the RBC suspension normalized by that of the suspending medium) as a function of shear rate at a hematocrit level of 45%. The predictions from the MS-RBC model aligned remarkably well with blood viscosity measurements obtained from three separate laboratories which are expressed in green and red color [61].

### 3.2. RBC Mechanics

To study how RBCs behave under different conditions, such as in stenosis, aneurysms, and heart diseases, simulations using the Discrete Element Method (DEM) were conducted. A deformable RBC model showed good accuracy in reproducing mechanical responses under static loadings. Dynamic calibration of RBC deformation and viscoelastic behavior also showed good agreement. The model, coupled with an immersed boundary approach, assessed the flow properties of a single RBC in plasma. These results support the design of microfluidic devices for biological purposes. Limitations include not considering the incompressibility of RBCs’ internal fluid and interactions among many RBCs. More detailed modeling could provide better insights into RBC mechanics [62].

Balogh and Bagchi’s work [63] focuses on complex geometry chips with non-moving vascular walls enclosing blood corpuscles and plasma. They use an immersed boundary method with FEM, a sharp-interface ghost-node method for static walls and moving objects, and a finite volume/spectral technique. Results align well with real-world cases, aiding in RBC flow, platelet interaction, cell sorting, and microfluidic device design. A microfluidic device with coflowing and micropillar channels measures RBC deformability. Lower blood pressure in the coflowing channel indicates reduced deformability, causing higher pressure to drop in the micropillar tube as shown in Figure 4. This device accurately gauges blood pressure and RBC deformability without expensive equipment [64].

A study simulated blood flow through microfluidic channels with varying numbers of RBCs and cylindrical obstacles. A machine learning algorithm predicted RBC position and velocity from video analysis. The study found microfluidic channels ideal for simulating RBC mechanics in laminar flow. Limitations included the machine learning analysis of real-time blood flow footage reprinted with permission from Elsevier [65].

For dielectrophoretic applications, a method for optimizing flow distribution in a conventional benchtop microfluidic chamber was developed in another study. In a diagnostic microenvironment, the separation of suspended micrometer-sized particles is crucial. Dielectrophoresis (DEP), a widely used separation technique in BioMEMS diagnostic applications [66,67], is frequently recommended for the physical manipulation and characterization of different types of cells and particles [68]. A design modeled after blood veins aims to improve DEP systems in labs, focusing on architecture and cell dispersion. Simulations show that two-level bifurcation results in uniform flow, while further bifurcation slows the flow and causes cell suspension deposition. Rounded corners are suggested to enhance homogeneous cell adherence. Physical characteristics of RBCs in different human diseases are described in Table 2 [69,70,71,72,73,74,75].

### 3.3. Modeling Blood as a Fluid with Deformable Cells Flow

To study the dynamics of cell movement, their deformation necessitates a challenge which is to represent the fluid flow in its entirety—fluid flow plus cells should be represented in simulation with frequent mesh adaptation. Together with cells that should follow a Lagrangian path, the front tracking method (FTM) is used to couple cells to fluid flow [76] or immersed boundary method (IBM) [77]. Additional algorithms have been developed to approximate fluid flow in a Eulerian framework (fixed mesh). Like the FTM, IBM asserts that the local fluid velocity determines the way cell vertices move while also applying a force to the fluid flow.

Owing to advection with the fluid motion of the solid structure, the IBM coupling approach may be rigid, limiting the time step in simulations. To match the local velocities of the fluid and cells, another technique incorporates viscous fluid flow–cell vertices coupling [78]. This coupling technique may cause a partial slip at the fluid–cell interface, affecting stability and time step. Monitoring fluid mesh nodes inside and outside a cell is crucial to differentiate internal and external fluids. Particle-based approaches use free particles to model these fluids, achieving separation through fluid particle bounce-backs off a moving membrane. The no-slip condition is maintained through collisions between fluid particles and membrane vertices in MPC [79] or viscous force coupling in DPD [80]. Accurate fluid flow resolution around deformable cells need a detailed grid or high particle density. The cell membrane network must be fine enough to model deformations. These factors are problem-specific and should be checked for refinement, typically assuming equivalent length-scale resolution for both fluid and membrane. Table 3 shows some blood cell dynamics visualization and measurement using microfluidics. 

## 4. Hematological Disease and Disorder Modeling

There are many diseases related to the blood flow system. They range from atherosclerosis, aneurysm, sickle cell disease, blood cancer, etc. Investigating the cause, mechanism, behavior of affected cells, and effects requires extensive research and experiments. With current computational power and the emergence of a wide variety of numerical techniques, CFD analysis has become an increasingly popular method for the detailed study and modeling of such disease scenarios.

### 4.1. Thrombosis Development Investigation

This is a very important part of the study and one requiring major attention, since after approximately half a century of increasing computational capacity and blood research (more than 10,000 publications on blood coagulation in PubMed), several labs have started tackling the problem of systems analysis of thrombosis (more than 350 publications on blood coagulation simulation in PubMed). To bring genomics, high-dimensional phenotyping, biomarker surveillance, imaging, and even more to life, systems biology aims to give medical practitioners treating patients an integrative framework. To do this, it uses microfluidic devices as modeling environments [82]. The role of microfluidics in hematological disease studies are in mentioned in a systematic cyclic process in the Figure 5 below.

A key component of arterial thrombosis induction is the von Willebrand factor (VWF). Kim and Ku’s study [83] demonstrated that platelet VWF is crucial for forming occlusive thrombosis under high shear conditions. Using various models, they found that mice lacking platelet VWF did not develop occlusive thrombosis, highlighting the essential role of platelet VWF in this process. This has been demonstrated in Figure 6.

The study used a design of experimental approach to evaluate three extrinsic design factors affecting occlusion time variability. It found that channel fabrication methods and collagen surface coating (with fibrillar collagen being better) significantly impacted occlusion time, while the choice of anticoagulant (citrate vs. heparin) did not. Devices were designed using COMSOL for shear rates of 500 s^−1^ (normal) and 6500 s^−1^ (stenosed) [85]. In earlier studies, a technique called Device Thrombogenicity Emulation was used to model microfluidic platforms that simulate the shear stress profiles of mechanical circulatory support (MCS) devices. These studies focused on replicating shear stresses in detail by using geometric variations of channels and computational fluid dynamics (CFDs) [84]. The formation of fibrin mesh using adapted microfluidic (eight parallel channels 250 µm wide by 60 µm tall generated in PDMS) surfaces that include significant triggers for coagulation (TF) and platelet aggregation (collagen) affects thrombus growth, stability, and embolization. It was reported that the growing clot can withstand shear stress exceeding 2000 dyne/cm^2^ due to a rise of 12 to 28 times in shear resistance provided by the fibrin network [86]. An even more extensive approach in designing flow chambers and adopting these in research especially using CFD simulations is given in this paper where diseases or disorders like von Willebrand disease, hemophilia, and congenital platelet disorders have been discussed [87]. 

Along with these, more on using microfluidic devices and their potential benefits were demonstrated [88,89] Michelle et al showed that our microsystem can function as an in vitro model of HUS and that, in the context of HUS, shear stress has an impact on microvascular thrombosis/obstruction and the medicine eptifibatide’s ability to lower platelet aggregation. CFD modeling indicated that wall shear stress/shear rate typical of physiological microvascular conditions was found at centerline flow velocities and viscosities mentioned in Figure 7 [89]. However, some limitations in this field of research were pointed out, for example, by Susan et al., who mentioned that low Re limits the ability of microfluidic systems to model secondary complex flows, separation, and pulsatile. Larger in vitro macro fluidic setups [90] may be needed to evaluate the impact of intricate flows and device surfaces on thrombosis, such as those conducted by Krajewski et al. who assessed the thrombogenicity of neurovascular stents using a recirculating loop.

### 4.2. Development of a Circulating Tumor Cell Detection Device 

The impact of physical restrictions on blood flow, such as studies on the effects of cardiovascular wall shear stress (WSS) on CTC bundles showed that rising WSS levels are linked to an increase in the disaggregation of the bundles in Figure 8 [90], portrayed below.

The most effective isolation techniques, according to Mary et al., are the conventional approach and immunocytochemical technologies, which label CTCs for separation based on unique surface antigens that distinguish them from normal bystander cells. A method utilizing biotin-tagged antibodies, which bind to specific surface antigens to differentiate them from healthy bystander cells, is outlined here. The antibodies are added to a suspension of blood cells with the aim of only displaying surface biotin molecules on CTCs. Subsequently, the cell solution is sent via a microfluidic channel that has around 9000 transverse posts coated with streptavidin [91].

## 5. Emergence of Experimental Microfluidics for Hemostasis Analysis

### 5.1. Microfluidics as a Key Soluton to Hemostasis Analysis

There is currently no way of performing an immediate hemostasis test, and as a result, clinical decisions are sometimes postponed for several hours [92]. A new design concept with functional integration of the microfluidic chip has the potential for point-of-care (POC) testing. A lot of work has recently gone into creating microfluidic flow cytometers that are compatible with extracorporeal circuits [93]. Ideally, these devices could be used by institutions for POC platelet analysis with minimal settings. They are straightforward to operate and require little in the way of user training [94]. Thus, there is a straightforward and dependable method for utilizing native blood for thrombosis monitoring and therapy at home and in hospitals. It is possible to simulate mechanical injury bleeding and see the interplay of various hemostasis components in vitro by combining a micro-engineered pneumatic valve with a microfluidic device [95]. Additionally, a microfabricated film bulk acoustic sensor can measure hemostatic parameters and monitor blood clotting in real-time [96]. A lateral electric field was used to excite a film stack of Au, ZnO, and Si_3_N_4_ to create the device. It revealed the sequential clotting stages by operating under a shear mode and recorded the alterations in resonance frequency in conjunction with the changes in blood viscosity.

### 5.2. Pulsatile Circulatory Fluid Flow and Mixing Device

In biological processes, pulsatile flows are crucial [97] in the cardiovascular system, which circulates pulsatile blood through the body. It is crucial to simulate and experiment with hemodynamic circumstances in pulsatile blood flow to comprehend cardiovascular disorders. Most of the research uses commercially available pumps or pressure controllers to create pulsatile or oscillatory flows on a microscale. The internal pressure regulation of the controller causes them to frequently create major departures from the ideal waveform, such as overshoots at abrupt pressure changes and longer reaction times. Therefore, an effective time variant on chip microfluidic flow control is necessary to mimic the biological microenvironment to eliminate reliance on external sources such as thermal [98,99], electrical [100], magnetic forces [101].

Apart from the pressure pumps, pulsatile flow can be generated using on-chip microfluidic devices [97,102] such as oscillators consisting of PDMS microvalves, mechanical capacitors, and fluidic resistance utilizing only the pressure head [103,104]. This most recent technique is very simple and straightforward to generate pulsatile flow and can be used for cardiovascular research due to its versatility in both flow rate and switching time of the valve to control the flow. Additionally, microfluidic oscillators are compatible with rapid prototyping from both 3D printing and photolithography with soft lithography at the end.

Microfluidic mixing devices can be divided into “active” and “passive” types [105]. Passive micromixers frequently use complicated channel geometry to promote diffusion or chaotic advection, while active micromixers typically require external energy sources. Passive micromixers only need energy to drive the fluids. Various micro-mixing techniques have been developed, broadly discussed, and analyzed in many studies mentioned in Table 4 [105,106].

### 5.3. Principles of Design of Microfluidic Tools in Coagulation Studies

Microfluidics supports methods for studying platelet aggregation mechanisms. Coagulation research can engineer hemodynamic characteristics and blood vessel damage, including thrombosis development, stenosis creation, microvasculature construction, and shear rate changes.

The pathophysiology of many hematological and cardiovascular diseases like sickle cell disease, bleeding disorders, and uncontrolled blood clotting in atherosclerosis or stroke is closely linked to the mechanical properties of blood cells and vascular tissues. [112,113]. Biomarkers for detecting hematologic and cardiovascular diseases can be derived from tissue and blood features like stiffness, platelet contraction forces, and blood vessel stiffness.

Blood clot elastic modulus helps in the study of platelet retraction forces. Increased contraction force is linked to clotting disorders like chest discomfort, coronary artery disease, and Buerger’s disease [114]. Measuring platelet contraction forces is challenging due to various factors. Blood disorders can affect platelet function differently. Atomic force microscopy (AFM) helps quantify these forces, with a modified side view approach measuring individual platelet contractile force and elasticity [115]. This method assesses platelet contraction rates at the individual cell level. PDMS micropillars, acting as springs, help monitor and calculate forces from platelet adhesion and contact [116,117]. Optical tweezers can also be used to evaluate the activation status and binding capacity of particular integrin–fibrinogen pairings of active platelets [118]. Similar methods can quantitatively study platelet mechanics in response to matrix changes, enhancing our understanding of hemostasis and thrombosis. Eventually, microfluidic device design principles for coagulation research are very important to know which roughly gives a through idea in Figure 9 [115,119,120,121].

### 5.4. Blood Sample Preparation

The fractionation of blood samples into their constituent parts is an initial stage in blood-related research. The process of separating red blood cells is often carried out using centrifugation. Although it is the most effective and easiest way to separate blood components, it has the potential to lyse blood cells and contaminate the sorted levels during extraction [122]. Flow cytometry is another popular method for sorting blood cells. However, flow cytometry is a laborious process that necessitates specialized equipment, trained scientists, and a lot of time to complete a single analysis [122,123]. Microfluidic devices are a relatively new technology that shows promise for clinical diagnostic point-of-care system integration, as well as for methods for separating blood components. Passive methods for cell separation in microfluidic devices have received a lot of attention because they use straightforward channel designs and flows driven by pressure to separate [122]. Filtration using membranes and filters made by micromachines that induce retention of cells is one passive method for blood plasma extraction [124,125]. Techniques like hydrodynamic filtration [126], pinched flow fractionation (PFF) [127], and deterministic lateral displacement (DLD) [128] have proven effective in separating RBCs and WBCs. A relatively new method that is quickly gaining traction is inertial microfluidics, which uses hydrodynamic forces to concentrate cells inside the flow, allowing for on-chip cell sorting and analysis [129,130].

### 5.5. Microchannel Fabrication

The science and technology of microfluidics is defined by a variety of micro-domain effects [131,132] caused by the minuscule size (1–1000 μm) of fluidic channels. These channels have various uses, including chemical separations, dynamic cell cultures with carefully controlled microenvironments, and point-of-care diagnostics [133,134].

Before designing a microchannel, it is important to understand fluidic resistance and capacitance in microvascular environments. The electrical–fluidic analogy is a widely used simple and quick method for determining it rather than the complex and time-consuming CFD simulation [103,135]. The most popular ones are photolithography, 3D printing, and soft lithography.

#### 5.5.1. Photolithography

A key method in microfluidic processing, photolithography has been widely used to create microstructures on substrates like silicon, glass, and quartz. Using mask lithography or electron beam exposure techniques, the photoresist is coated onto silicon wafers to create microfluidic chips. Its main idea is to manipulate the UV light to take advantage of the chemical characteristics of photoresists and produce exact microstructures and microchannels [136,137]. Developing the chip pattern, making the mask, exposing the chip, etching, and cleaning are the main processes in photolithography. A schematic of the photolithographic chip manufacturing process is shown in Figure 10.

#### 5.5.2. 3D Printing and Soft Lithography

The allure of fabricating microfluidic devices with 3D printing [139,140] stems from its unique properties. (1) Complex 3D geometries can be easily fabricated with 3D printing [141]. (2) Rapid prototyping made possible by 3D printing makes it simple to fine-tune. (3) These devices can then mimic the physicochemical and biochemical features of real organs [142]. In contrast to soft lithography [143], 3D printing creates microfluidic devices that are robust enough to respond to high flow rates and high pressures [131]. When compared to soft lithography [144], above all, the steadily increasing resolution, rough surfaces created by 3D printers, and biocompatibility of materials pave the way for the creation of microfluidic devices with actual micro-scale channel dimensions (≤20 μm). PDMS is extensively used as a structural material for making channels from molding; however, hydrophobicity, chemical swelling, and shrinkage limit its application with biomolecules.

Another method, 3D bioprinting, an emerging bio-fabrication technique, deposits tissues by adding biomaterials layer by layer. Its main advantages are cost-effectiveness, versatility, and time efficiency. This technology can recreate the 3D structure of the vascular network and efficiently construct various models [145]. Currently, various techniques are employed in bioprinting to create 3D perfusable vascular channels in vitro, such as extrusion bioprinting, inkjet bioprinting, and light-assisted bioprinting.

Laser degradation uses laser energy to selectively degrade cell-filled hydrogel, forming a network structure. Endothelial cells grow within this hydrogel to create a 3D vascular network. This method is high-resolution, easy to use, and maintains sterility [146]. It works with materials like hyaluronic acid, collagen, PEG, and agarose, allowing precise control of fluid channels and flow.

There are other methods for making microchannels for vascular flow, such as self-assembly, whose template is described in the literature [138], and the thermal air expansion method for creating semicircular channels for arterial thrombosis investigation [147]. Most recently, pressure-responsive circular microchannels mimicking blood vessels [148] based on PDMS and ECM-based microchannels [149] have been described.

### 5.6. Surface Functioning of Microfluidic Device for Thrombosis Experiment

#### 5.6.1. Conformal Coating

High shear rates can cause ECM proteins to be flushed away, leading to instability and variation in protein coverage. This issue can be resolved by seeding the coated ECM protein onto silanized glass (treated with 3-aminopropyl triethoxysilane) [150]. The findings show that silanized glass is a stable and suitable medium for cell culture. Interactions at blood–material interfaces in microdevices are crucial for treating thrombotic diseases. Despite advancements in low-activating coatings, device function is often limited by humoral responses and cellular adhesion [151]. Biological surface coatings, like endothelial cells (ECs), help prevent adverse material-related reactions and enable continuous anticoagulation. Understanding material–blood interactions is crucial for creating hemocompatible coatings. Key factors include wettability, functional groups, and topographical roughness [152].

#### 5.6.2. Microcontact Printing

It is possible to accurately reproduce the specific thrombotic areas by means of microcontact printing. On a micro or nanoscale, the process is analogous to stamps and inkpads. The method is used to covalently bind human fibrinogen in micron-sized, randomly placed islands with different regional coverages, as shown in Figure 11b, which is a simple microcontact printing strategy [153]. It is not possible to create fibrin that is perpendicular to the flow with an enzyme concentration that is too low. In the relevant pathophysiological setting, printed microarrays enable various TFs and proteins to assemble on the microfluidic channel, allowing for the development of purpose-specific cues for platelet regulation [154,155].

#### 5.6.3. Ultraviolet Photolithography

In biomedical applications, ultraviolet (UV) photolithography is a commonly used microfabrication process. Spots of TF are shaped into microcapillary flow models using deep-UV photolithography as seen in Figure 11c. It is possible to see the connection between shear rate and the start of coagulation [156]. Since cell signaling and reactions could be easily investigated, microfluidic live-cell microarrays have great promise for a variety of biological research applications [157]. UV photolithography ensures the accuracy and geometry of microarrays. Using microarrays, platelet interaction with spatial and physical cues in the microenvironment can be investigated [158]. Moreover, it is simple to investigate how various protein microarrays or microtopographies affect platelet behavior [159,160]. Though UV photolithography is a complicated procedure requiring certain skills, the uniformity and reproducibility of micro/nanopatterning are guaranteed.

## 6. Realtime Monitoring of Thrombus Formation and Antithrombotic Medication Assessment

### 6.1. MEMS Realtime Monitoring of Thrombus Formation

The self-reporting vascular graft [161] shown below in Figure 12, has the benefit of providing a more direct assessment of the occlusion’s size and nature, which was previously unattainable. This will enhance the process of clinical decision-making, while also providing the potential advantages and opportunities of community-based healthcare monitoring. The integration of electrical impedance spectroscopy with an implanted radiotelemetry device allows for the monitoring of grafts. This has the potential to be used for timely identification of venous stenosis and thrombus development in real-time inside a living organism. The approach has the potential for use for various cardiovascular illnesses and vascular implanted devices. 

Microfluidic tools for examining the mechanics of blood clots are details described in the previously reported works mentioned in the Table 5 [162,163,164,165].

### 6.2. Whole Blood Antithrombotic Medication and Thrombosis Assessment

This microfluidic system combines tortuosity-driven gradients in the microdevice with shear gradients caused by stenosis, allowing blood clots to form more quickly and with less blood volume [166]. Preliminary results suggest that this tortuosity-activated microfluidic device could aid in diagnosing clotting issues and guiding thrombotic or antithrombotic treatments. The device mimics stenosed tortuous arterioles, creating zones of sudden fluid acceleration (pre-stenosed), non-uniform shear and tortuosity (stenosed + tortuous), and sudden deceleration (post-stenosed) (Figure 13a). The fluid experiences multiple stages of acceleration and deceleration along the device, as demonstrated by our CFD research in Figure 13b,c. As demonstrated in their earlier study (Figure 13d), they also discovered that the wall shear stress rapidly varies at the pre- and post-tortuous regions [167]. Moreover, the absolute wall shear rate and its gradients increased with the inflow velocity, indicating that higher flow rates may favor blood clot formation. Consequently, to enhance blood clotting, this design’s three distinct shear gradient zones—pre-stenosed, stenosed + tortuous, and post-stenosed—all cooperate. The apparatus is subsequently constructed, placed under a microscope for visualization, and linked to a syringe pump to initiate flow (Figure 13e). The 4-channel architecture is somewhat comparable to an in vivo vascular network where clots may develop or separate locally, yet systemic thrombosis causes the pressure to still rise. For added convenience, the device’s overall length and width were made to fit on a typical glass microscope slide (Figure 13f). The surface of the microchannels was functionalized inside the device with collagen type I, a frequently used platelet agonist [168], to facilitate faster cell activation and adherence (Figure 13g). Using fluorescence imaging, they were able to observe the production of varied-sized thrombi along its entire length (Figure 13h).

For antithrombotic treatment research, it is becoming essential to know the recent advancements in microfluidic instruments described in some previous works in Table 6 [169,170,171,172,173].

## 7. Critical Appraisal of Disease Study Approach Using Microfluidics

It seems that for broadly applicable investigations, the use of CFD software has provided significant improvement in our perception and understanding of cardiovascular flow and related disease mechanisms [174]. These include but are not limited to blood flow dynamics from single phase to multiphase spectra involving cell–cell interactions in addition to diseases like stenosis and tumor cell circulation. However, for more complicated issues or novel research in this field, researchers have relied more on actual testing without any prior CFD analysis. These devices have also been manufactured using calculations from medical data, again without any sort of optimizations in the design phase using fluid dynamics modeling software. While these have been the cases till now, a very common practice has been using CFDs for initial analysis or gathering information at important points demanding attention in a microfluidic device and then designing devices relying on these results instead of trial-and-error methods of experimentation as software give more dedicated and robust [175] information. Additionally, using the obtained data in CFD software for the final comparison of results has been achieved. The point to be noted here is that it seems that it is possible that using a microfluidic device environment that replicates the original vascular system greatly simplifies the geometry and study, thus the use of general-purpose CFD software easily enables carrying out convoluted vascular research. However, there is still no exact replacement for direct experimental evidence that can only be obtained by making the device and testing it under in vitro or in vivo conditions [176]. The best approach of course would be to utilize both and exchange data back and forth to make the final observation as robust and reliable as possible.

Despite the significant advancements in microfluidic technology for vascular flow and pathological investigations, several challenges remain. One of the primary obstacles is the fabrication complexity and reproducibility of microfluidic devices, which can limit their scalability and widespread adoption. Achieving precise control over the microenvironment to mimic physiological conditions accurately is another critical challenge, as even minor deviations can affect experimental outcomes. Moreover, the integration of computational fluid dynamic (CFD) simulations with experimental data requires high computational resources and sophisticated modeling techniques, which can be resource-intensive and time-consuming [177]. Ensuring biocompatibility and long-term stability of materials used in microfluidic devices also poses significant hurdles, especially for in vivo applications. Additionally, there is a need for standardized protocols and validation methods to ensure consistency and reliability across different studies. Addressing these challenges is crucial for the continued progress and practical application of microfluidic technology in vascular research and pathology.

In the near future, microfluidic devices can be tailored to individual patients, allowing for personalized drug testing and treatment optimization. Mimicking patient-specific blood flow conditions will enable precise evaluation of antithrombotic therapies. Automation and miniaturization in microfluidics facilitate high-throughput screening of potential drug candidates [178]. Researchers can rapidly test multiple compounds, accelerating drug discovery for thrombosis prevention and treatment. Integrating microfluidic platforms with advanced imaging techniques (such as confocal microscopy or optical coherence tomography) provides real-time visualization of thrombus formation. This synergy enhances our understanding of clot dynamics and responses to interventions.

Combining experimental microfluidic data with computational fluid dynamic (CFD) simulations allows us to explore multiscale phenomena. These models can predict thrombus formation at different vessel sizes, aiding in the design of targeted therapies [179]. The accuracy of simulations depends on several factors, including patient-specific geometry, the chosen CFD methodology, blood properties, and flow rates at both the inlet and outlet [46]. These elements are crucial for ensuring that the simulations accurately reflect real-world conditions and provide reliable results. Future challenges include assessing the consistency of flow patterns across different methods to ensure reliable CFD results. Further investigations are also important to enhance the reliability of hemodynamic simulations, considering the variability and sensitivity of CFD results due to various pre-processing steps and assumptions [180]. Researchers are working toward integrating CFD and microfluidic devices into animal models for in vivo validation. These hybrid systems will provide a more realistic environment, validating findings obtained from in vitro studies.

Emerging trends indicate a shift towards integrating machine learning and artificial intelligence with microfluidic systems to enhance vascular models’ precision and predictive capabilities. Future clinical applications could include predicting sites of cardiovascular events, such as the formation or rupture of atherosclerotic plaques, aneurysms, and thrombosis. To fully realize this potential, large, prospective, image-based clinical studies are necessary to evaluate the effectiveness of biomechanical parameters in predicting well-defined clinical endpoints [181]. This integration promises to enable real-time monitoring and adaptive control of microfluidic environments, paving the way for more accurate and personalized medical interventions. The considerable differences in patient geometry and boundary conditions complicate the training of data-driven models in high-dimensional feature spaces. Enhancing fast and real-time CFD and FEA simulations with ML algorithms could improve the clinical applicability of these biomechanical tools [182]. Additionally, the development of more sophisticated, multi-organ-on-chip platforms could revolutionize drug testing and disease modeling, providing deeper insights into complex vascular diseases and their systemic impacts. These devices’ continued miniaturization and automation are expected to improve their accessibility and usability in clinical settings, potentially transforming diagnostic and therapeutic approaches in vascular medicine.

Microfluidics bridges disciplines such as biology, engineering, and medicine. Collaborations between experts in these fields will drive innovation and accelerate the translation of research findings into clinical practice. Efforts to standardize microfluidic protocols and ensure reproducibility are crucial. Establishing guidelines for device fabrication, cell culture, and data analysis will enhance the reliability of results. In summary, microfluidic technology holds immense promise for advancing our understanding of arterial thrombosis. As researchers continue to explore these prospects, we anticipate breakthroughs that will transform thrombosis management and patient care. Ongoing collaborations between mechanical engineers and clinical and medical scientists are crucial. CFDs can play a significant role in understanding the pathophysiology of cardiovascular disease and developing innovative treatment methods in the cardiovascular field.

## 8. Conclusions

The science of cardiovascular mechanics is still continuously developing and requires highly sophisticated devices to test theories. Theories regarding the flow of blood as a single-phase fluid, multiphase with different cells and plasma suspension, and even the particles or cells exclusively are almost non-exhaustive, and findings are many. These findings might at times disagree but still provide insights to make the field even more mature and precise. The main aim of our paper is to focus on the emerging technology of microfluidics that have helped accelerate complicated blood flow and related disease research at a never-before-seen rate. It seems that there are numerous theories of vascular flow, diseases, and disorders in biofluids like blood in developing microfluidic models or devices. These theory-based devices for inquiring more into the root causes of emerging vascular flow problems have been studied and attempts have been made to compile and cover them as extensively as possible within our scope. The promising future of CFDs can be seen in studying multiscale problems as algorithms and methods are becoming more and more efficient. However, the results of a simulation depend heavily on the physics involved and a deep understanding of it. Often, incorrect assumptions of a supposedly insignificant parameter might lead to deviated results because that parameter might be important for the case at hand. Hence, relying on CFD alone might not be a good call to make even if the study is novel. Again, experimental testing at the first iteration might not give the desired results too but repeated tests are time-consuming and costly, not to mention there are limited resources as well. Thus, directly designing a device might not be the recommended method for successful lab-on-a-chip development. In this paper, we have tried to highlight many of the diseases that are common and rampant in the world. Addressing them and quickly setting up labs or equipment is a must. To achieve this and to propel new research, knowledge of the current state-of-the-art facilities in the software sector, as well as instruments and devices have been described here, with their limitations whenever possible. Another crucial part is choosing the right method to fabricate and manufacture this device. Of note is that most of the studies mentioned here have focused on the efficiency and accuracy of measurement of different parameters in the blood to understand and scrutinize diseases and abnormalities, not on the economic aspects or on comprehensive mass production strategies. This review article also includes, while discussing the disease and their study methodology, ways in which they were produced. These manufacturing methods are themselves cutting-edge technologies and demand their own elaborate discussion in case of application to microfluidic devices. However, this was outside the scope of this paper. The application of CFDs and experimental modus operandi in critical evaluation and investigation of cases using microfluidic technologies as the basis were presented here. An attempt was made to answer the question of whether CFD alone, fabrication and use of devices, or a combination of both would give the best results. It seems that though the use of both would be best, there are certain use cases where the first two alone would suffice to answer critical questions, considering time and cost.

## Figures and Tables

**Figure 1 sensors-24-05872-f001:**
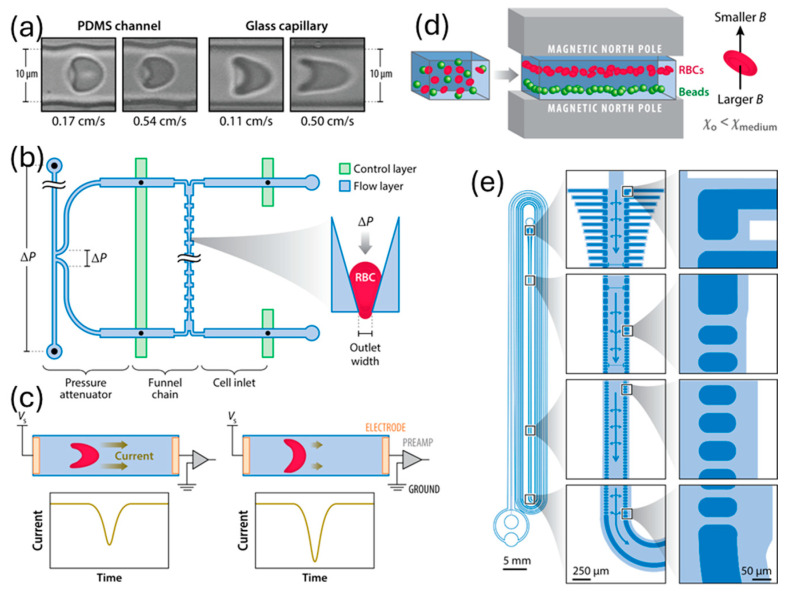
(**a**) Red blood cells deform in PDMS channels and glass capillaries [36]. (**b**) Red blood cells (RBCs) undergo deformability-dependent passage through microchannel constrictions at specified pressures. The microfluidic device can identify RBCs infected with malaria at different phases of the parasite’s growth by applying pressures that are finely controlled using an on-chip attenuation system [40,41]. (**c**) Electrical opacity to evaluate the deformation of RBCs inside small capillaries [42]. (**d**) A magnetic medium allows for the simultaneous sorting of cells and analysis depending on density. This technique has been easily integrated into a point-of-care microfluidic platform and may identify abnormalities in red blood cell density [43]. (**e**) Controlled incremental filtration could passively sort WBCs continuously by capitalizing on their margination dynamics [44]. Common abbreviations used in this context include B for magnetic field strength, RBCs for red blood cells, PDMS for polydimethylsiloxane, WBCs for white blood cells, and χ for magnetic susceptibility. Reprinted from ref. [45].

**Figure 2 sensors-24-05872-f002:**
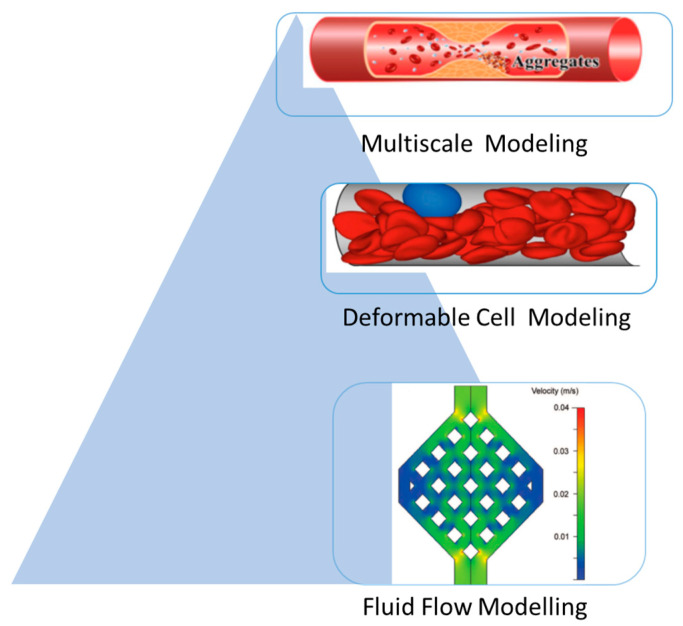
Modeling scales of blood flow in a vascular system.

**Figure 3 sensors-24-05872-f003:**
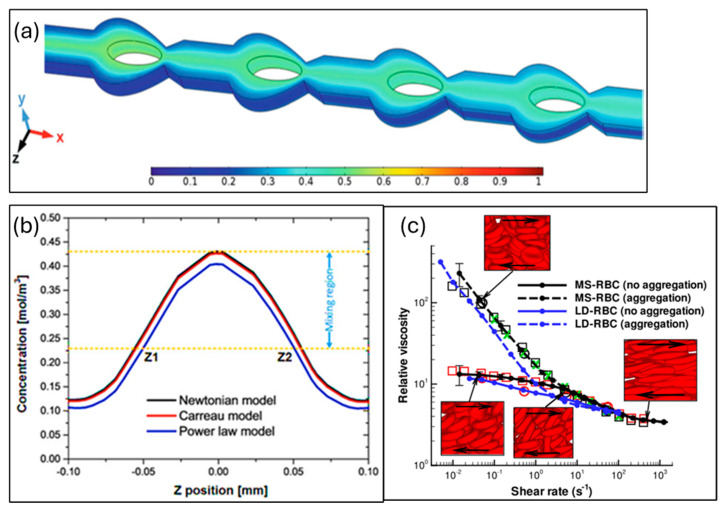
(**a**) Illustration of fluid mixing in an SAR mixer. (**b**) A contrast of the power law, Carreau, and Newtonian models’ concentration profiles at the outlet: input velocity = 0.05 mL/min. Reprinted from ref. [59]. (**c**) Numerical result validation for whole blood with shear rate as independent and Ringer ES. Non-Newtonian relative viscosity as the dependent variable [61].

**Figure 4 sensors-24-05872-f004:**
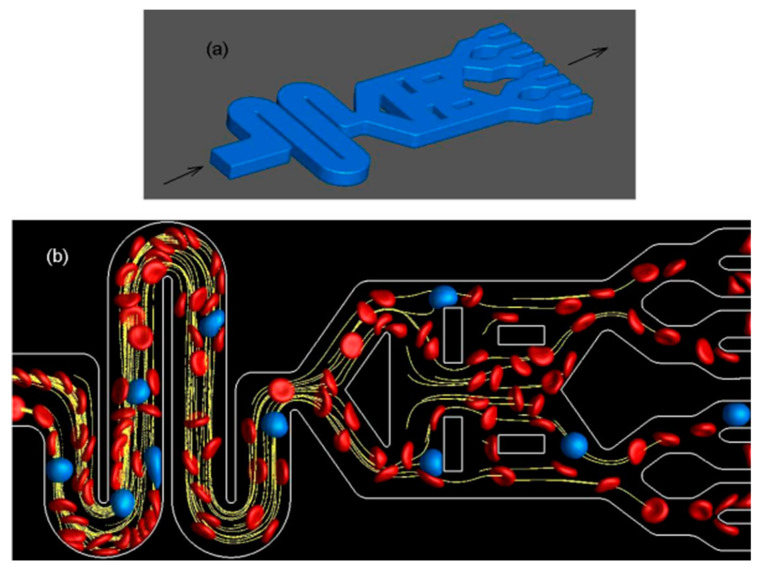
CFD analysis of the flow of suspended, deformable RBCs and WBC/CTCs in a lab-on-chip device that is geometrically complex (RBC—Red, WBC—Blue, others—yellow). (**a**) Device geometry. (**b**) Blood cell visualization as particle tracking along the flow channel. Adapted from [63].

**Figure 5 sensors-24-05872-f005:**
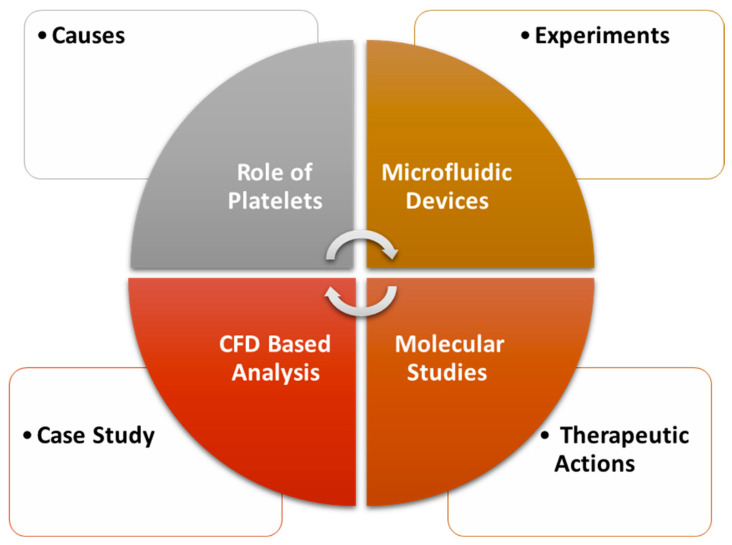
Role of microfluidics in hematological disease studies.

**Figure 6 sensors-24-05872-f006:**
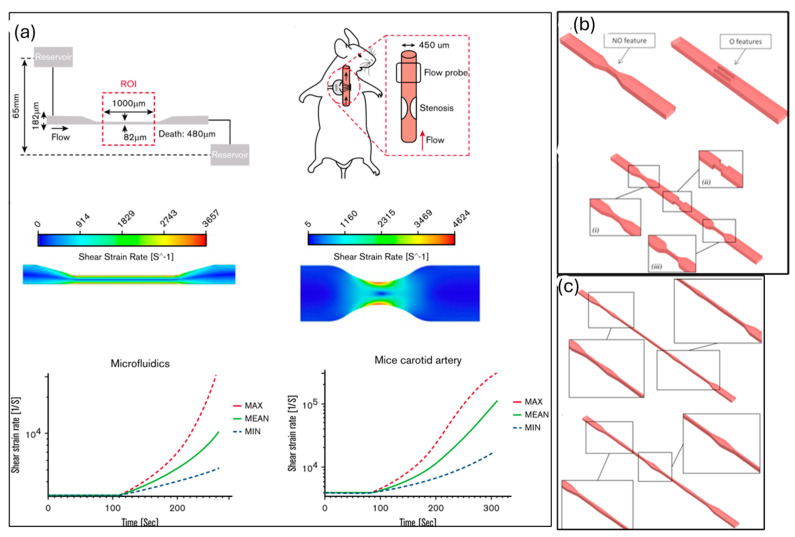
(**a**) Characteristics of hydrodynamic flow through a stenosed mouse artery and a high-shear microfluidic device, adapted from ref. [83]. (**b**) An illustration of obstructive (right) and non-obstructive (left) geometries (i) angled, (ii) abrupt and sharp, and (iii) smoothed. (**c**) Enlarged views of the three distinct types of narrowing—angled, abrupt and sharp, and smoothed—are displayed in subpanels. Reprinted with permission from Springer [84].

**Figure 7 sensors-24-05872-f007:**
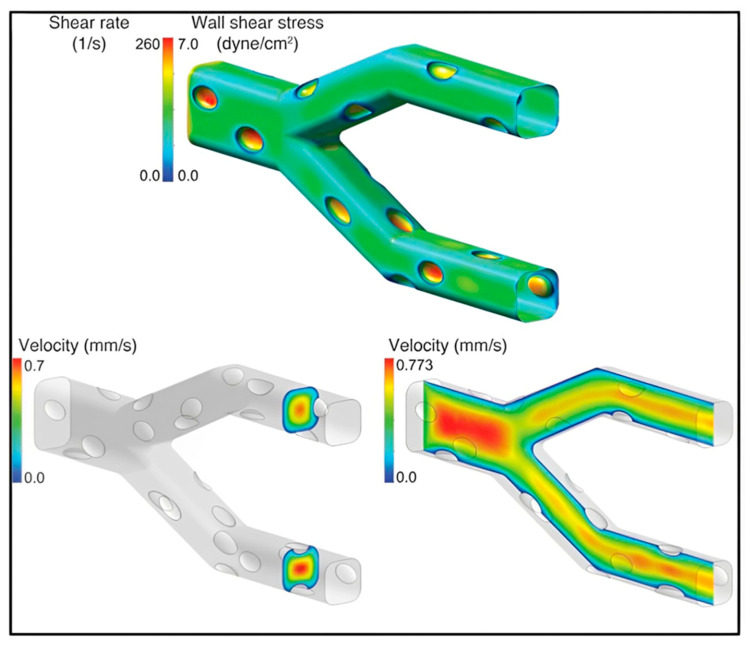
This image shows the “endothelialized” microvasculature on a chip. CFD modeling indicated that wall shear stress/shear rate typical of physiological microvascular conditions was found at centerline flow velocities and viscosities. Reprinted with permission from ASCI [89].

**Figure 8 sensors-24-05872-f008:**
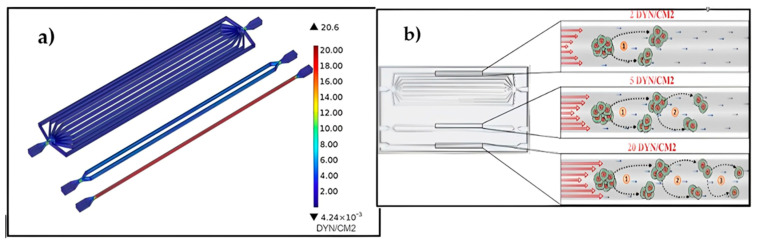
Diagram showing the effects of increasing SS levels on CTC cluster disaggregation. (**a**) SS flow profiles for the flow dynamics and SS profiles of the channels’ fluid at a 30 mL/min inflow. (**b**) SS causes the disaggregation of CTC clusters as observed in the reported work [90].

**Figure 9 sensors-24-05872-f009:**
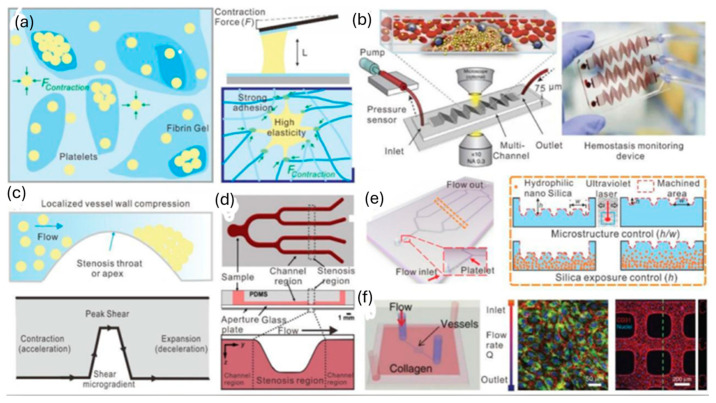
Microfluidic device design principles for coagulation research [119]. (**a**) Atomic force microscopy (AFM) calculation of individual platelet contractions [115], (**b**) hemostasis monitor device and method, (**c**) diagram showing the three primary parts of the shear micro gradient or stenosis, (**d**) schematic of microfluidic chip stenosis, (**e**) schematic and surface micropatterning of a device for detecting platelet aggregation, adapted from [120], (**f**) channel vessel endothelial cells and collagen gel micro-vessel system schematic, adapted from [121]; reprinted with permission.

**Figure 10 sensors-24-05872-f010:**
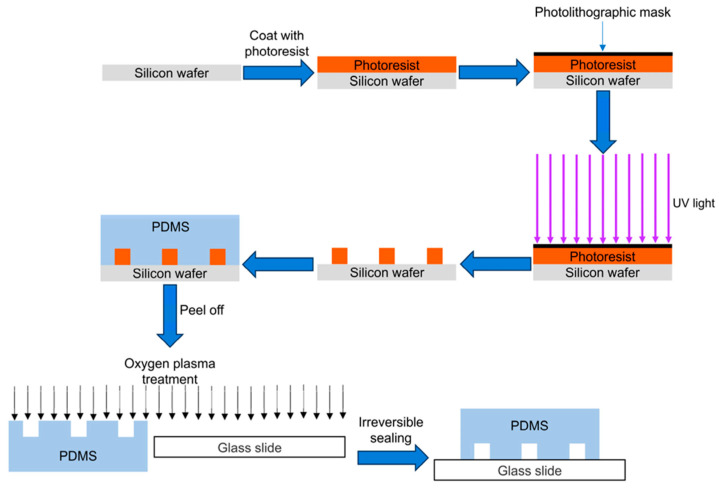
Photolithography and soft lithography processes for microchannel fabrication, reprinted with permission from ref. [138].

**Figure 11 sensors-24-05872-f011:**
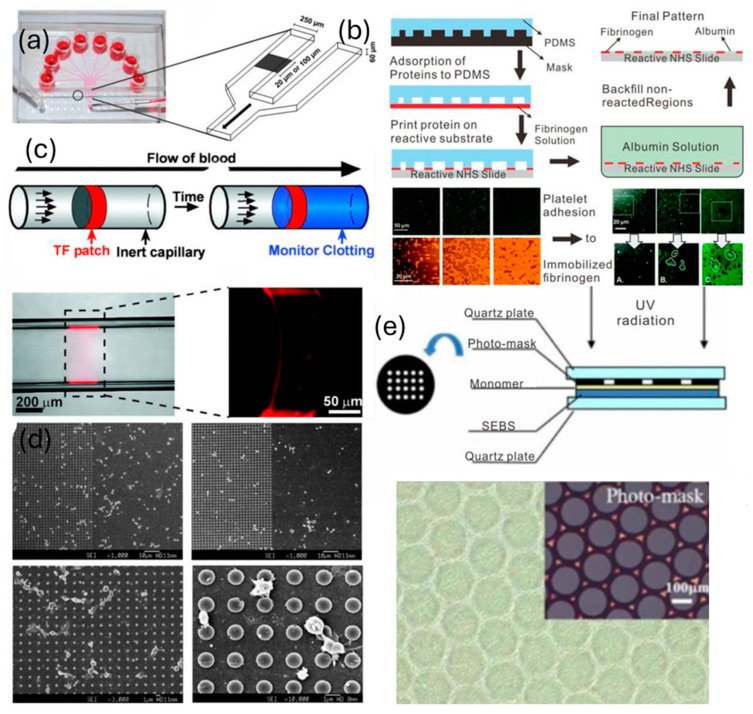
Surface engineering using micro/nanotechnologies in microdevices (**a**) A microdevice with eight channels with a stripe of protein that runs normal to two adjacent channels; (**b**) microcontact protein printing schema for assessing platelet adhesion and morphology with covalently immobilized fibrinogen and albumin; (**c**) the microcapillary flow models were created using patches of tissue factor (TF) using deep-ultraviolet (UV) photolithography; (**d**) using UV photolithography, platelets interacted with various microarray interspacing; (**e**) patterned (SEBS) styrene-block-(ethylene-co-butylene)-block-styrene exposed to ultraviolet light reveals surface structure and adhesive sites of platelets; reprinted with permission from ref. [119].

**Figure 12 sensors-24-05872-f012:**
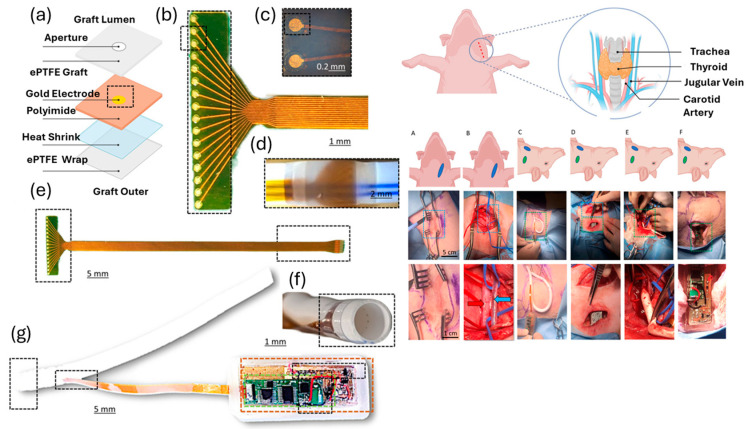
(**a**) Diagram illustrating the layers involved in graft creation; (**b**) 16 gold electrode arrays made of polyimide; (**c**) an enlarged view of the ×10 sensor array; (**d**) in situ electrode array, intraluminal lit to activate the sensor and sensing apertures; (**e**) complete array highlighting the distal telemetry connection and the proximal sensor; (**f**) a prototype device with transparent thermal PTFE shrink to secure the sensor and intraluminal circumferential sensor holes on a single plane inside the graft; (**g**) prototype gadgets connected to the power supply (highlighted in gold), wireless impedance electronics (highlighted in green), and magnetic reed switches used in in vivo research; reprinted with permission from ref. [161].

**Figure 13 sensors-24-05872-f013:**
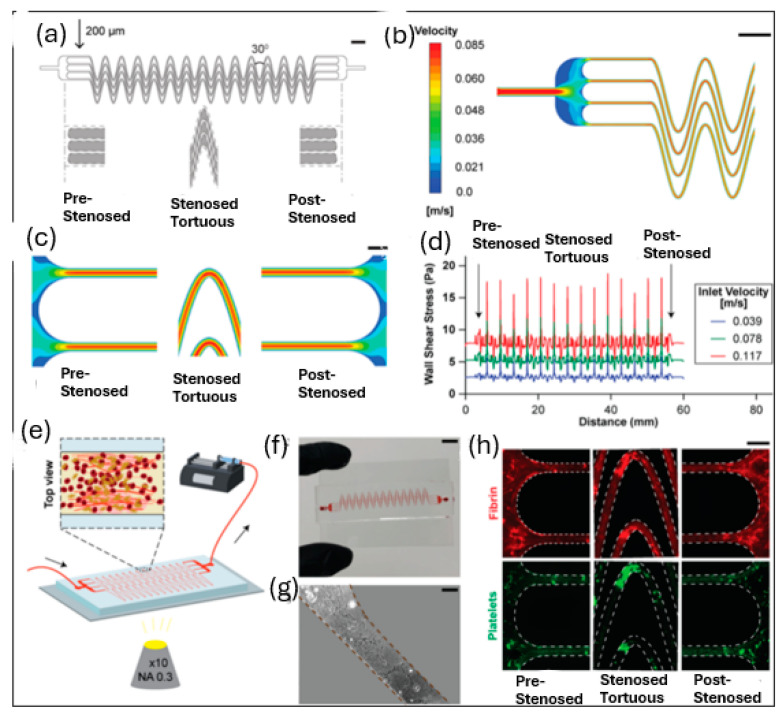
(**a**) The design includes three regions: accelerating pre-stenosed, uniform stenosed tortuous, and decelerating post-stenosed. (**b**) A heat map shows velocity distribution. (**c**) Images depict the regions at an inlet velocity of 0.078 m/s. (**d**) A graph illustrates wall shear stress along the device length at various inlet velocities. (**e**) A schematic shows the device on a microscope connected to a blood pump. (**f**) A photograph of the device after an experiment. (**g**) An image of rat tail collagen type I coating a section of the device. (**h**) Fluorescent micrographs display blood clots: fibrin (red) and adhered platelets (green) in the pre-tortuous, stenosed tortuous, and post-tortuous regions [166].

**Table 1 sensors-24-05872-t001:** Composition, shape, size, and density of all components of Blood.

Component	Shape	Diameter (µm)	Percentage	Viscosity (mPa·s)	Density (kg/m^3^)	References
Whole blood	Liquid	-	-	4–5 (37 °C)	~1055	[25]
Plasma	Liquid	-	~53%	1.10–1.35 (37 °C)	~1021	[25,26]
Platelets	Oval discoid	2	~1%	-	~1060	[27]
WBC	Spherical	6–20	~1%	-	~1090	[25]
RBC	Biconcave disk	8	~45%	6–7	~1100	[25,27]

**Table 2 sensors-24-05872-t002:** Physical characteristics of RBCs in different human diseases.

Disease	Rigidity	Membrane Integrity	Size	Viscosity	Dehydration	Stiffness	Deformability	References
Sickle cell disease (SCD)	Increased	-	Decreased	Increased	Increased	Increased	Decreased	[69,70]
Malaria	Increased	Altered	Decreased	Increased	-	Increased	Decreased (1.5–200 Times)	[45,71,72]
End-stage kidney disease (ESKD)	Increased	-	Decreased	Increased	-	Increased	Decreased	[73]
Diabetes (II) mellitus	Increased	Altered	No Significant Change	-	Increased	Increased	Decreased	[69,74]
Thalassemia	Increased	Altered	-	-	Increased	-	Decreased	[72]
Hypercholesterolemia	Increased	Altered	-	-	-	Increased	Decreased	[75]

**Table 3 sensors-24-05872-t003:** Blood cell dynamics visualization and measurement using microfluidics. Reprinted with permission from ref. [81].

Microfluidic Technique	Cell Types	Calculating Deformability Degree	Homogenous Flow	Cell Measuring Capability	Precision	Limitations
Channel–Fluid-induced Cell Deformation	RBCs	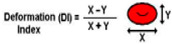	Yes (but not the extensional flow)	Large amounts in a single run	Good	Costly micro-visualization apparatus
Channel–Fluid-induced Cell Deformation	RBCs and WBCs	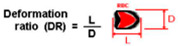	No	Large amounts in a single run	High sensitivity, precise control	Costly micro-visualization apparatus,
Channel–Fluid-induced Cell Deformation	RBCs and WBCs	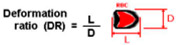	No	Large amounts in a single run	Capacity to differentiate healthy and diseased cells	Costly micro-visualization apparatus and in vitro experimental results may be needed for numerical model validation
Channel–Fluid-induced Cell Deformation	RBCs	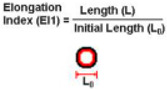	Yes (but not the extensional flow)	Large amounts in a single run	Capacity to differentiate healthy and diseased cells	Costly micro-visualization apparatus
Channel–Structure-induced Cell Deformation	RBCs	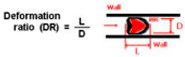	Yes (but not the extensional flow)	-	Capacity to differentiate healthy and diseased cells	Flow control proves to be complicated; difficult fabrication; probability of blockage; costly micro-visualization apparatus

**Table 4 sensors-24-05872-t004:** A short review of various flow mixing devices.

Mixer Type	Fluid Rheology	Flow Regime	Shape	Image	Fabrication Chemical	Advantages	Ref.
Passive	Newtonian (glycerol) viscoelastic (PAM solution)	Pulse	T-Shaped	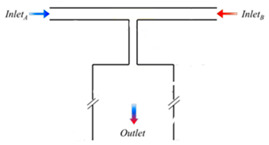	PDMS	Low cost Ease of fabrication	[107]
Active, numerical	Viscoelastic (bovine blood)	Laminar	Novel asymmetric network	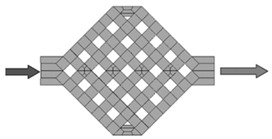	PDMS	Biocompatibility Low cost Easy to mold Optical clarity	[108]
Passive	Viscoelastic	Laminar	Microchannel	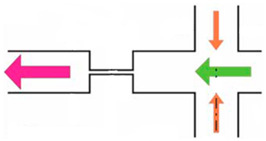	Silicon/glass chip	-	[109]
Passive	Viscoelastic (PAM Solution) Newtonian (Glycerol)	Pulse	T-Shaped	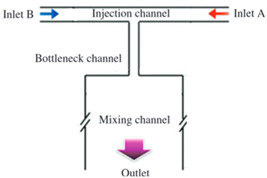	PDMS	Biocompatibility Low cost Easy fabrication Optical clarity	[110]
Active (induced charge electro-osmosis)	Polyacrylamide	-	T-Shaped	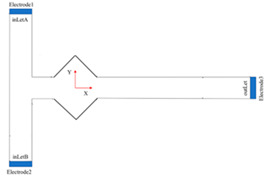	rheoTool (Finite volume method)	Low cost	[111]

**Table 5 sensors-24-05872-t005:** Microfluidic tools for examining the mechanics of blood clots.

Model	Microfluidic Size	Anticoagulant	Detection Principle	Results	Ref.
Actuated surface-attached posts (ASAPs)	d = 200 µm	Citrate	Utilise actuated surface-attached post (ASAP) technology to gauge blood clot stiffness throughout the formation process.	verified that the ASAP system can measure the hardness of blood clots.	[162]
A silicon master	Master:15 µm × 15 µm × 25 µm, Posts: 4 µm in diameter and 15 µm in height	Heparin or sodium citrate	platelet force during clot retraction measured by microscale column deflection	By measuring platelet force, one can determine which trauma patients may need hemostatic intervention and quantify the platelet response to various activators.	[163]
Microclot array elastometry (clotMAT) system	w = 1189 µm, d = 498 µm (narrow size: w = 132.7 ± 17.2 µm, d = 80.7 ± 5.9 µm)	Citrated PRP	Platelet force can be used to assess the response of platelets to different activators and identify trauma patients who may require hemostatic intervention.	simulates the development of clot mechanical characteristics and shape under both normal and aberrant coagulation circumstances.	[164]
8-channel microfluidic device	w = 250 µm, d = 60 µm	CTI	Keep an eye on platelet deposition and record dynamic thrombus formation in the presence of venous shear.	The occlusivity and blood flow resistance of a clot can be separately improved by the surface density and exposure area of the tissue factor.	[165]

**Table 6 sensors-24-05872-t006:** Microfluidic instruments for antithrombotic treatment research.

Model	Detection Principle	Microfluidic Size	Method	Anticoagulant	Results	Ref.
Parallel plate flow chamber	To ascertain whether PC and VWF can help individuals receiving clopidogrel regain their ability to coagulate	d = 120 µm, w = 450 mm, l = 2 cm		Heparin	Although clopidogrel’s effects could not be reversed by PC or VWF, their combination resulted in a small improvement.	[169]
8-channel microfluidic device	Identification of platelet sensitivity to inhibitors of P2Y1, COX-1, and P2Y12	d = 60 µm, w = 250 µm	Standard photolithography	PPACK	In vitro, treating P2Y12 and P2Y1 inhibitors simultaneously is more successful than treating antagonists alone.	[170]
Microcontact printing	Antiplatelet medication effects on platelet aggregation and adhesion after stenosis	d = 0.18 mm, w = 1 mm, l = 70 mm (narrow size: w = 0.2 or 0.4 mm)	-	PPACK	Aspirin cannot reduce platelet adhesion, GPIIb/IIIa inhibitors significantly reduce adhesion to fibrinogen	[154]
8-channel microfluidic device	Evaluation of the status of platelet aggregation following the administration of aspirin	w = 250 µm	Standard photolithography	PPACK	When added in vitro, ASA decreases platelet deposition and stops the rate of artery wall shear.	[171]
The Maastricht flow chamber	Sensitivity of thrombosis to antiplatelet and contractile drugs at physiological temperatures	d = 50 µm, w = 3 mm, l = 3 mm	Microspot-based high-throughput technology	Sodium citrate	When ASA is introduced in vitro, the rate of arterial wall shear is stopped, and platelet deposition is reduced.	[172]
8-channel microfluidic device	measures the rate of platelet and fibrin aggregation and mimics the hemophilia A/B model by blocking the activity of coagulation factors in the blood.	250 µm × 250 µm	Standard photolithography	CTI	On collagen/TF, anti-FVIII slightly inhibited fibrin production, while on collagen/FXIa, it totally stopped fibrin generation.	[173]

## Data Availability

No new data were created or analyzed in this study. Data sharing is not applicable to this article.

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
