# Peer review of "A Synergistic Overview between Microfluidics and Numerical Research for Vascular Flow and Pathological Investigations"

_sensors, 2024, doi:10.3390/s24185872_

Round 1

Reviewer 1 Report

Comments and Suggestions for Authors

This manuscript provides a comprehensive overview of the use of microfluidic devices to investigate the rheological properties of blood, the forces associated with cardiovascular diseases, experimental models and methodologies, device fabrication, and the applications of these studies. It also discusses how these findings can guide clinical practice and direct future research. Overall, the paper offers insightful explanations of why a combination of computational fluid dynamics (CFD) and experimental approaches can provide more detailed information on disease mechanisms, which are replicated on a microfluidic platform to mimic the original biological system, and aid in the development of the device or chip itself.

However, the manuscript in its current form requires substantial revisions to be considered for acceptance. 

1)The manuscript's length is excessive; authors should revise and streamline the content, aiming to condense the page count to 30 pages in alignment with scholarly article conventions. The paper must concentrate on pivotal discoveries and theoretical contributions, eschewing extensive preliminaries and tangential details.

2)The number of figures in the manuscript is excessive, with many providing minimal information. Authors should reduce the figures to a maximum of 10 key diagrams that present clear, straightforward information directly relevant to the text. They should consider consolidating or eliminating figures with low information yield to enhance the efficiency of information conveyance.

3)It is suggested that the authors update the content , particularly on the topics of multiphysics coupling, the application of deep learning in CFD, and the role of Physics-Informed Neural Networks in solving Partial Differential Equations. These cutting-edge technologies are current research hotspots and warrant thorough discussion.

4)The manuscript requires enhanced critical analysis. A review should critically assess existing research, clearly identifying issues and challenges within the field. For example, authors should explore the limitations of microfluidic technology in practical applications, including costs, reproducibility, and clinical translation barriers.

5)The manuscript should bolster research on the practical application and implementation of microfluidic technology, especially in medicine and pathogen detection. Authors are advised to delve into specific application cases of microfluidic technology in clinical and public health settings, providing detailed use examples. Case studies illustrating the devices' roles in clinical practice and diagnostics, with examples demonstrating their impact on medical services, would aid readers in understanding the practical value and clinical implications of these technologies.

6)The number of references in the manuscript is too large; authors are advised to streamline to a maximum of 200 references, ensuring that only the most relevant and current studies are included. Removing those with little or outdated relevance will improve the literature review's quality and pertinence.

Comments on the Quality of English Language

The authors need to understand the distinction between writing a paper and writing a book.

Reviewer 2 Report

Comments and Suggestions for Authors

This review focus on advances in microfluidic technology and research for vascular flow and pathological investigations, in which I think the manuscript is well organized and written and I recommend the acceptance of this review after the following issues are solved. Also please focus on the point 6, try to improve it emphatically.

1. In Table 1, the references 29-31 need to be indicated separately in every row. The current version is unclear.

2. Some tables are not indicated in the text, please revised them.

3. The figures also need to be indicated in the text, please revised them.

4. There are too many figures in this review and I think that some of them may be combined into one figure. Such as figure 21 and 22.

5. Table 3 is missing the related references.

6. This review is missing the important perspective part, and the personal point of view from the authors, which need to be greatly improved.

7. There are not only the photolithography technique and 3D printing could be used to fabricate the microchannels, please make more reviews for this part.

8.  The title is too long and not highlight the key points of the topic thus I recommend to improve it. 

Round 2

Reviewer 2 Report

Comments and Suggestions for Authors

All my previous concerns have been reply well by the authors.